# Momentum-dependent power law measured in an interacting quantum wire beyond the Luttinger limit

Y. Jin[1], O. Tsyplyatyev[2], M. Moreno[1], A. Anthore[3], W.K. Tan[1], J.P. Griffiths[1], I. Farrer [1,6], D.A. Ritchie [1], L.I. Glazman[4], A.J. Schofield[5] & C.J.B. Ford [1]

Power laws in physics have until now always been associated with a scale invariance originating from the absence of a length scale. Recently, an emergent invariance even in the presence of a length scale has been predicted by the newly-developed nonlinear-Luttinger-liquid theory for a one-dimensional (1D) quantum fluid at finite energy and momentum, at which the particle's wavelength provides the length scale. We present experimental evidence for this new type of power law in the spectral function of interacting electrons in a quantum wire using a transport-spectroscopy technique. The observed momentum dependence of the power law in the high-energy region matches the theoretical predictions, supporting not only the 1D theory of interacting particles beyond the linear regime but also the existence of a new type of universality that emerges at finite energy and momentum.

[1] Department of Physics, Cavendish Laboratory, University of Cambridge, Cambridge CB3 0HE, UK. [2] Institut für Theoretische Physik, Universität Frankfurt, Max-von-Laue Straße 1, 60438 Frankfurt, Germany. [3] Université Paris Diderot, Sorbonne Paris Cité, 75013 Paris, France. [4] Departments of Physics and Applied Physics, Yale University, New Haven, CT 06520, USA. [5] School of Physics and Astronomy, University of Birmingham, Edgbaston, Birmingham B15 2TT, UK. [6] Present address: Department of Electronic & Electrical Engineering, University of Sheffield, 3 Solly Street, Sheffield, S1 4DE, UK. Correspondence and requests for materials should be addressed to O.T. (email: o.tsyplyatyev@gmail.com) or to C.J.B.F. (email: cjbf@cam.ac.uk)

Power laws play an important role in physics and they are generally associated with a scale invariance originating from the absence of a length scale. The most notable example is in continuous phase transitions, where the diverging correlation length means that microscopic details become irrelevant and universality classes characterise the exponents[1–6]. Response functions associated with the dynamics of manifestly scale-invariant soft excitations in a quantum system, such as X-ray absorption in a metal[7–10] or the spectral function of a Tomonaga–Luttinger liquid (TLL)[11–15], likewise display power laws around zero momentum. Recently, the possibility of invariance emerging even in the presence of a length scale was predicted by the newly developed nonlinear Luttinger-liquid theory for a one-dimensional (1D) quantum fluid at finite energy and momentum[16]. Here, the length scale is determined by the particle's wavelength.

In 1D, the effects of electron–electron interactions are amplified strongly, structuring free electrons into collective excitations. These excitations are charge and spin density waves within the TLL theory, which approximates the electron dispersion relation with a linear energy–momentum dependence. The TLL spectral function (which gives the probability of finding an electron with a particular energy and momentum) is zero at energies below the dispersion of the collective mode, and follows a power law in energy measured from the threshold defined by that spectrum; the corresponding exponent depends on the interaction strength[17,18]. To go beyond the linear approximation of the energy–momentum dependence and take into account the parabolicity of the dispersion of free electrons, the mobile-impurity model was developed[19], leading to a nonlinear hydrodynamic theory that extends the low-energy universality of a TLL to finite energy, corresponding to excitations from far below the Fermi energy to just above it. The exponent becomes momentum dependent through a finite curvature of the spectral-edge dispersion that changes with the momentum, defining the momentum dependence as a unique feature of the nonlinear hydrodynamics in 1D. For electrons (fermions with spin 1/2), this dispersion is close to parabolic[20] and the mobile-impurity model with spin and charge degrees of freedom predicts an essential dependence of the threshold exponent on momentum away from the Fermi points[21].

Here, we probe the electron dispersion and threshold exponents experimentally by measuring the momentum- and energy-resolved tunnelling between neighbouring 1D and 2D systems, formed within the two quantum wells of a GaAs/AlGaAs double-well heterostructure. We find experimental evidence for the new type of power law in the spectral function. As we probe the energies below the bottom of the 1D dispersion, the tunnelling current drops away more slowly than that predicted by a non-interacting model and cannot be explained by a power law with a momentum-independent exponent. This excess conductance is instead consistent with the momentum-dependent exponent predicted by the new mobile-impurity model described above[20,21]. This measurement supports not only the 1D theory of interacting particles beyond the linear regime, but also the existence of a new type of universality that emerges at finite energy and momentum. The result is a significant stepping stone towards a systematic understanding of a wider variety of many-body systems, ranging from quantum optics to high-energy and solid-state physics.

## Results

**Principle of experiment.** Our devices contain two closely spaced two-dimensional electron gases (2DEGs). The upper 2DEG is depleted into 1D channels by Schottky gates. A small current may flow between the two layers due to quantum tunnelling (in the $z$-direction). In order for tunnelling to occur, some filled states of one system must have the same momentum and energy as some empty states of the other, as momentum and energy are conserved. In order to explore the entire dispersions of each of the systems, a DC voltage $V_{DC}$ can be applied between the layers, providing extra (or less) energy for tunnelling. The dispersions can be thought of as being offset in energy one from the other. Likewise, a magnetic field $B$ applied in the plane of the layers causes a Lorentz force which boosts the momentum of electrons as they tunnel between the layers. When this in-plane $B$ is perpendicular to a 1D wire (in the $y$-direction), the resultant momentum change is along the wire (in the $x$-direction). In a 2D system, the dispersion is a paraboloid in energy as a function of momentum. The corresponding density of states in energy and momentum is called the spectral function.

In Fig. 1a and b, spectral functions of two parallel 2D systems are shown, to illustrate the offsets in momentum $\hbar k_x$ (caused by $B$ in the $y$-direction) for which significant tunnelling occurs. Here, there is no energy offset ($V_{DC} = 0$), so tunnelling occurs at the Fermi energy $E_F$ (where there are both empty and filled states). In Fig. 1a, $B$ is quite small, so the paraboloids touch on the inside, probing the Fermi surface at $k_x = -k_F$, where $k_F$ is the Fermi wave vector. In Fig. 1b, $B$ has increased until the paraboloids touch on the outside, probing the Fermi surface at $k_x = k_F$. The green lines mark the states contributing most to the tunnelling. If $V_{DC}$ is non-zero, empty states at the Fermi energy of one paraboloid overlap filled states of the other (or vice versa) away

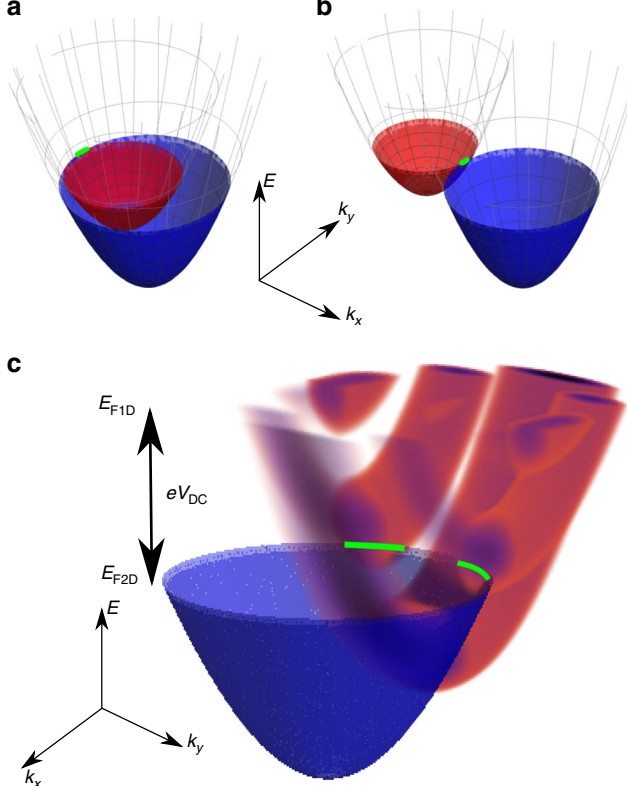

**Fig. 1** Spectral functions. **a, b** Overlap of the spectral functions of two 2D layers as a function of momentum ($k_x, k_y$) and energy $E$—current only flows from occupied states to empty states where the spectral function is not negligible. Magnetic field $B$ displaces one paraboloid to the right. Fermi circles touch on the inside/outside, probing states of the red paraboloid near $-k_F/k_F$ at the Fermi energy $E_F$. **c** A 2D system (blue) probing multiple subbands of a 1D system (red), for finite $V_{DC}$

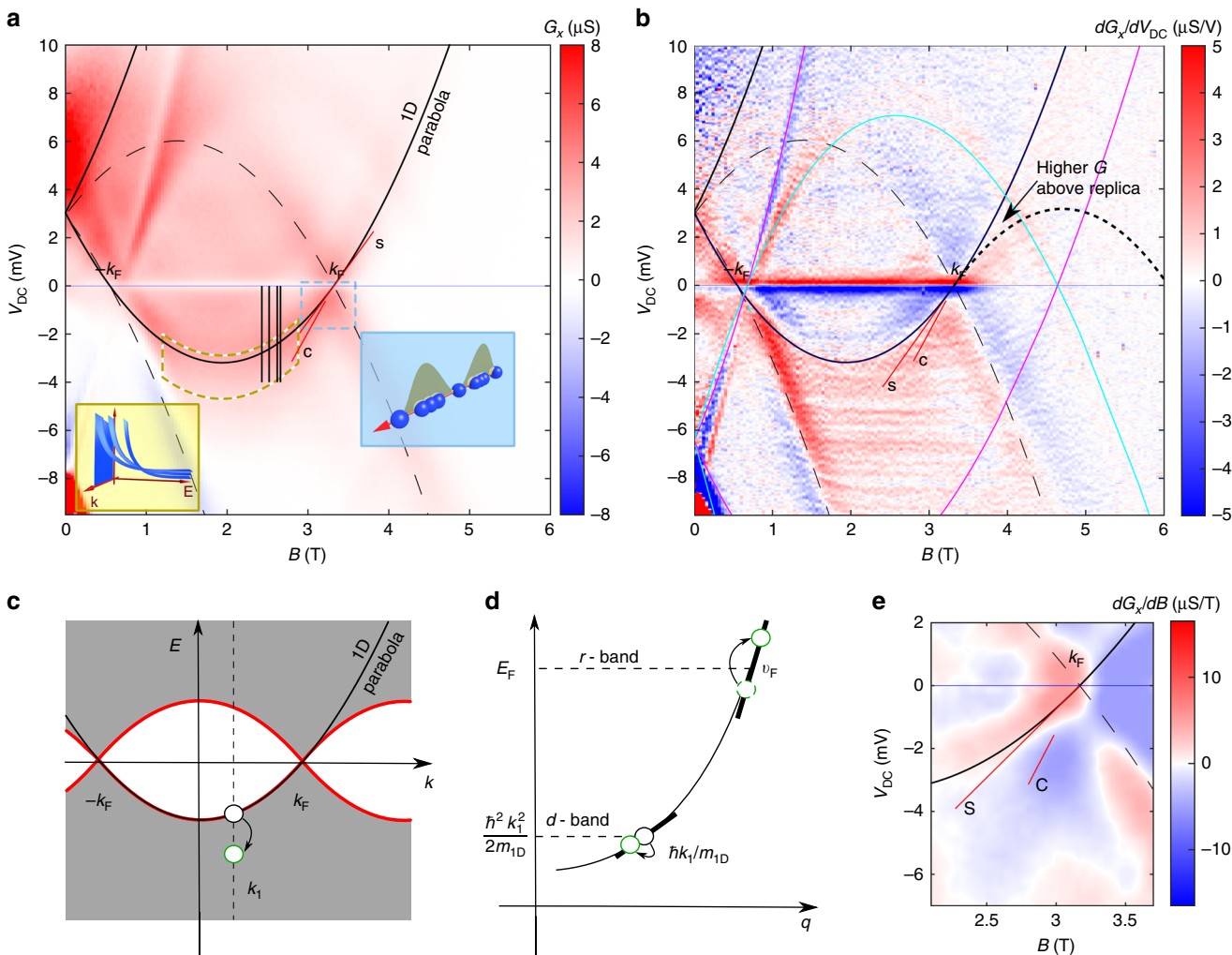

**Fig. 2** Conductance maps of spectra, and types of excitation. **a** Overview of conductance $G = dI/dV$ vs $B$ ($\propto$ momentum) and voltage $V_{DC}$ ($\propto$ energy $eV_{DC}$) (Sample A). The region along which $G$ is enhanced is shown enclosed by a dashed yellow line, and the left inset shows how $G$ might decay with energy, differently for each momentum $k$. Vertical black lines show cuts along which fitting is carried out. The region where spin–charge separation is visible is shown enclosed by a dashed blue line; the right inset illustrates charge (dots) and spin (shaded) waves. The bottom edge of the yellow region is selected as the line above which the nLL model still gives a reasonable fit, see Fig. 4. **b** Differential $dG/dV_{DC}$ of the raw data in (**a**), showing a replica above $k_F$ (where the conductance stays higher than expected for the non-interacting model in the region labelled Higher $G$, and then appears to drop off along a parabola that is an inverted replica of the main 1D parabola), separate spin and charge lines near $V_{DC} = 0$, labelled S and C, respectively and the parasitic signal (near magenta and cyan lines). **c** Dispersion of an interacting 1D system. The kinematically forbidden region is shown in white (see explanation in text), and grey indicates the continuum of many-body excitations. The thick red line is the border between the two regions. States on the border correspond to removing a single particle (black circle at $k_1$) from the many-particle state, and a higher-energy excitation described by the mobile-impurity model is marked by a green circle. **d** Splitting of fermionic dispersion into two subbands, one for a heavy hole with velocity $\hbar k_1/m_{1D}$ and one for excitations around $E_F$ with velocity $v_F$. Green circles are constituent parts of many-body excitations in the nonlinear regime (see explanation in text). **e** $dG/dB$ around the spin (S) and charge (C) lines. Their slopes indicate a spin-wave velocity of $v_s = 1.32 \times 10^5\,\text{ms}^{-1}$ and charge velocity of $v_c = 1.73 \times 10^5\,\text{ms}^{-1}$, respectively, with error bars of ~10%, see Supplementary Table 1

from $E_F$. In Fig. 1c, the 2D system is shown probing the more complex 1D spectral function (red) with multiple 1D subbands, for finite $V_{DC}$. The second 1D subband, for example, is split into two peaks in $k_y$, matching the Fourier transform of the corresponding spatial wave function.

The device therefore behaves as a spectrometer, using the 2D system to probe the spectral function of the 1D system (and vice versa). Figure 2a shows an overview of such a measurement, where conductance through the sample is measured as a function of energy ($\propto V_{DC}$) and momentum ($\propto B$). The conductance peaks form a set of intersecting parabolae, which correspond to the dispersions of each system. The parabolae corresponding to the 1D (2D) system are shown as solid (dashed) lines.

**Design of the nanostructure**. Our spectrometer devices are made with an MBE-grown GaAs/Al$_{0.33}$Ga$_{0.67}$As heterostructure with two parallel quantum wells 100 nm beneath the surface, both 18-nm wide, separated by a 14-nm tunnel barrier, giving a 32-nm centre-to-centre distance $d$. Figure 3a is an illustration of the device structure. An array of identical fine-feature wire gates (labelled WG) is fabricated on a Hall bar by electron-beam lithography. The gates are all joined together by air bridges, which are used to supply a negative voltage to deplete the upper 2DEG layer into 1D channels while leaving the lower 2DEG undisturbed. The use of air bridges allows the wires to be almost uniform along their whole length, rather than becoming narrower at one end, as would happen if all the gates were joined together

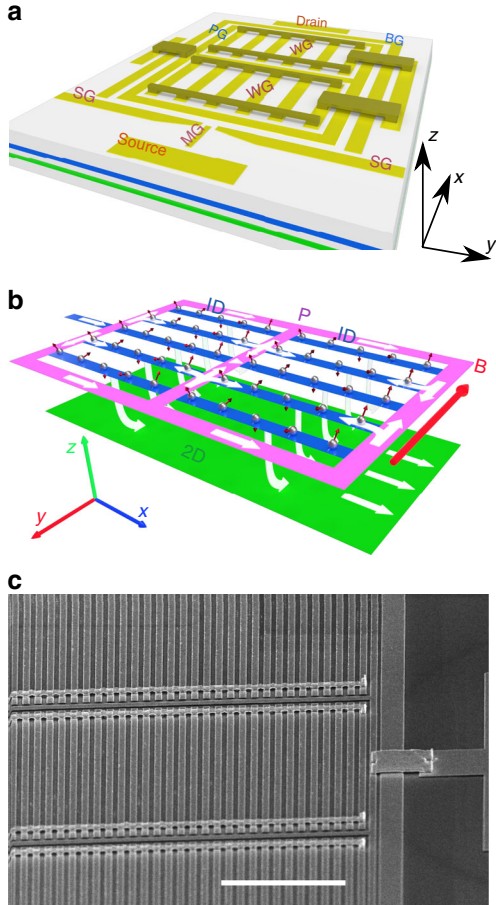

**Fig. 3** The 1D–2D spectrometer device. **a** Illustration of the 1D–2D spectrometer device. The control gates are split gate (SG) and mid-gate (MG) (depleting the lower layer but not the top, to inject current into the upper layer), gates defining 1D wires (WG), gate over the parasitic region (PG), and barrier gate (BG) confining the upper layer. Gates are not drawn to scale. In reality, they consist of a three-column array of over 500 wires, each of length 18 μm and gate width 0.1 μm, with 0.18-μm separation (device A) and 10 μm and gate width 0.3 μm, with 0.2-μm separation (device B). **b** Schematic of current flow: current enters the top layer under mid-gate MG at the left, flows into 1D wires (blue) via parasitic regions (magenta, labelled P) and tunnels to a lower, 2D, layer and out to a contact at the right. **c** Scanning electron micrograph of a device, showing the air bridges. The scale bar is 5-μm long

at one end by a gate on the surface, which would give a much greater electric field at the 2DEG than does a gate separated by an air gap. Details of our air-bridge technique will be published elsewhere (Y.J., J.P.G. and C.J.B.F., manuscript in preparation). A split pair of gates (labelled SG) and a middle gate (labelled MG) are positioned near the source contact of the Hall bar. The split gates are used to pinch off both the upper and lower 2DEG layers, while the mid-gate supplies a positive voltage to induce a channel in the upper 2DEG from where electrons can flow into the array. The split-gate/mid-gate combination ensures that electrons from the source Ohmic contact may only enter the 1D array via the upper layer. Current flows along a narrow strip from this injector to the entrance to each wire. This strip is labelled p in Fig. 3b as it provides an unwanted (parasitic) region in which tunnelling can occur. To control this region, it is covered by a gate (labelled PG in Fig. 3a). This allows the carrier density there to be varied, which changes the position of the parasitic tunnelling parabolae shown in magenta and cyan in Fig. 2b, but it has little effect on

their magnitude. On the opposite side of the device, near the drain contact, there is a barrier gate labelled BG, which pinches off just the upper 2DEG, so that electrons may only leave via the lower 2DEG, which leads to the drain Ohmic contact. As illustrated by white arrows in Fig. 3b, electrons must tunnel between the 1D wires and the lower 2DEG in order to travel between the source and drain contacts.

There are therefore two regions where electron tunnelling occurs from the upper well: (1) the 1D channels defined by the wire-gate array (shown in blue in Fig. 3b), and (2) the parasitic regions surrounding the wire-gate array, which allow electron flow into the array (shown in purple). We detect tunnelling from both regions in the experiment. Analysis is focused on the tunnelling from the wire array. The strength of confinement of the 1D channels can be controlled by the wire gates. The array, which contains 500 repeating units, provides a large total tunnelling area and hence a large-enough conductance to be measured with low noise. Tunnelling from the second region is parasitic—it cannot be eliminated due to device design. In order to remove the parasitic conductance contribution, a matching set of data is measured under identical conditions, but with the 1D wires just past pinchoff, such that the data contain only conductance due to parasitic tunnelling. After subtraction of these data from the conductance with the wires open, sharp features of the p-region (the magenta and cyan parabolae) are still noticeable, owing to a slight difference in the carrier densities, but in the regions of interest for the fitting to models (see later), the variation with density or field is slow, and so subtracting this background should be acceptable, though we always take into account the possibility that the parasitic contribution may have been scaled up or down by changing the effective area of the parasitic region with the wire gate.

The experiment was carried out at $T < 100$ mK in a $^3$He/$^4$He dilution refrigerator (sample A) or at $T \sim 330$ mK in a $^3$He cryostat (sample B). Device conductance was measured in a two-terminal phase-sensitive setup, where a small AC voltage was applied as the source–drain bias and the current response measured by a lock-in amplifier. The wire-gate voltage is chosen to be negative enough that only the 1D states in the lowest subband are populated, though for smaller voltages, up to three 1D subbands can be observed clearly. The conductance across the sample was measured as the DC bias was swept and the magnetic field incremented.

**Nonlinear phenomena.** The Luttinger model is only applicable in a small range of excitation energies about the Fermi energy $E_F$, which we define as the energy of the highest-occupied electron state relative to the bottom of the 1D subband. This corresponds to $V_{DC} = 0$. As we can measure excitations at all energies, including far above and below $E_F$, the predictions of the new nonlinear TLL model can be tested. We have previously observed signs of another nonlinear theory, a hierarchy of modes, which predicts the so-called replicas of the 1D dispersion at higher momentum and inverted in energy, shown as red lines in Fig. 2c[16,22]. We predicted the spectral strengths of the collective excitations forming the many-body continuum in the nonlinear regime to be inversely proportional to integer powers of the system length, separating the excitations into different levels of a hierarchical structure[22]. We indeed observed signs of the $k > k_F$ replica just above the magnetic field labelled $k_F$ in Fig. 2b[22,23], where the linear TLL model predicts exactly zero for the density of states. The extra excitations cause $G$ to remain high and then drop off quite rapidly along an inverted parabola below the 1D parabola at $V_{DC} > 0$, and this shows up as a broad red curve where $dG/dV_{DC} > 0$, inside the labelled region in the figure. This replica

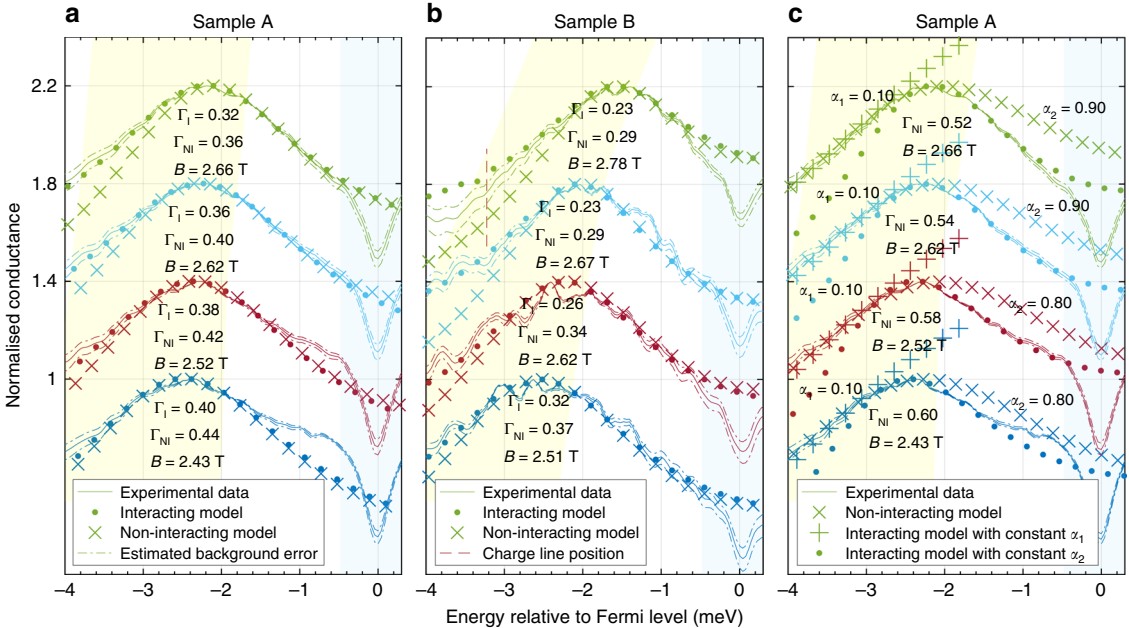

**Fig. 4** Fitting results showing the interaction-induced enhancement to the left of the peak. Fits to the conductance for (**a**) sample A and (**b**) sample B, normalised to their peak values and shifted vertically for clarity, for model 1 (crosses) and model 2 (dots). The dashed lines show the confidence interval of the measurement data within estimated error of the background conductance subtraction. The broadening fitting parameters for the non-interacting and interacting models ($\Gamma_{NI}$ and $\Gamma_I$, respectively) are shown, in units of millielectronvolts, together with the $B$ field at which the data were taken. Sample A (wires 18-μm long) was measured at 50 mK in a $He^3/He^4$ dilution refrigerator, while sample B (10-μm long) was measured at 330 mK in a $He^3$ cryostat. **c** Fits using other theories, or matching different parts of the data using different constant exponents $\alpha_1$ and $\alpha_2$, as labelled (using data for sample A)

parabola is expected to become indistinct further along its length, as observed. In shorter wires, an order of magnitude shorter than in this experiment, signs of a full replica were observed[24] in the principal region—the red line between the $\pm k_F$ points for $E > E_F$ (above the axis) in Fig. 2c, where the spectral strength of even the strongest many-body excitations is expected to be weaker by a factor proportional to the inverse square of the length of the wire, compared with the excitations forming the principal parabola.

In this paper, we turn our attention to the region below the bottom of the 1D parabola, and make detailed fits of the data to models. As we will show in Fig. 4a and b, we find that the tunnelling conductance is significantly enhanced over that predicted by the non-interacting model and cannot be fitted by a model with a simple, momentum-independent power-law dependence, in agreement with the new mobile-impurity model.

**Nonlinear models**. The region of interest in this paper (shown shaded yellow in Fig. 2a) is below the bottom of the 1D dispersion, where a momentum-dependent power law is predicted[25]. In the region of interest, the parasitic contribution to the conductance mentioned above varies very slowly with $B$, so the slight variation in density is negligible. The data with the parasitic contribution removed are therefore compared with calculations based on three models that differ in the form of the 1D spectral function: (1) without the power law (or other effect) arising from interactions, (2) with a momentum-dependent power law from interactions and (3) with a momentum-independent power law.

Model 1 is the full non-interacting model, and it contains a single parameter $\Gamma_{NI}$, the width of the disorder-broadened spectral function. The mobile-impurity model for electrons with Coulomb interactions (model 2) predicts the threshold exponents in terms of the curvature of the spinon mode in the nonlinear regime[21]. The spectral function of a single 1D subband in the hole sector, $A_1(k_x, E) \propto 1/|E - \varepsilon(k_x)|^{\alpha(k_x)}$, is measured in our experiment. For

a parabolic dispersion[20], $\varepsilon(k_x) = \hbar^2(k_x - k_F)^2/(2m_{1D}K_s)$, the exponent is

$$\alpha(k_x) = 1 - \frac{(1 - C(k_x))^2}{4K_c} - \frac{K_c(1 - D(k_x))^2}{4}, \qquad (1)$$

(see details of the calculation in Supplementary Note 2) where the momentum-dependent parameters are $C(k) = (k^2 - k_F^2)/(k^2/K_s - k_F^2 K_s/K_c^2)$ and $D(k) = (k - k_F)(k_F/K_c^2 + k/K_s^2)/(k^2/K_s^2 - k_F^2/K_c^2)$, and $K_s$ and $K_c$ are the usual Luttinger parameters. By renormalisation-group arguments, $K_s = 1$ in our experiment[26] and we use $K_c < 1$ as a fitting parameter in both the linear and nonlinear regimes. In addition to these interaction effects, we also include the effect of disorder-induced broadening (of width $\Gamma_I$), which smears the threshold singularities to become maxima of the spectral-function energy dependence. Model 3 uses a simple power law for comparison, where the dependence on momentum and $K_c$ is replaced with a constant-valued $\alpha$, so that the exponent is momentum independent.

**Fitting in the nonlinear regime**. The models are evaluated as functions of $eV_{DC}$ (which is equivalent to the energy relative to $E_F$) and $B$. Since the calculation is too time-consuming for automated fitting, we calculate the conductance for a range of values of each parameter, and compare the model to the experimental data visually in order to determine the parameters that best fit the experiment (Fig. 4). In Fig. 4, calculations of the 1D–2D tunnelling conductance using the interacting and non-interacting models are compared with our experimental data for two samples (A and B). Cuts through the data are shown as curves for several magnetic fields (marked with black lines in Fig. 2a). The data and calculations are normalised by their maximal values, which occur on the 1D parabola. The nonlinear regime is a relatively wide region at higher negative biases than the peak position (up to the left-hand edge of the yellow region in

Figs. 4, 2a), which corresponds to the grey continuum of many-body excitations in Fig. 2c. Since the spectral functions of the disorder-broadened interacting 1D system and the 2DEG are convolved, there is also a finite conductance to the right of the peak. The calculated conductance from a model must therefore fit the measured conductance over a wide range of energies on either side of the peak, including the yellow shaded region in the figure down to a large negative energy that is significantly further from the peak than the broadening $\Gamma$ (at least down to $E - E_F = -4.0$ meV), and the white part up to the blue shaded region ($E - E_F > -0.5$ meV) on the right (which marks the zero-bias anomaly, ZBA, to be described later). The models ignore any second subband (visible at low fields for sample A as an enhancement close to zero bias, about 2 meV away from the first subband).

The value of $K_c < 1$ affects the skewness of the conduction peak. We find that $K_c = 0.70 \pm 0.03$ best matches the conductance peaks for both samples and all fields. Figure 5a shows the range of acceptable $\Gamma_I$ values vs $B$, for model 2. Between 2.5 and 2.7 T, $\Gamma_I$ is at a minimum and is roughly constant, as here we avoid the 2D parabola, as well as the localised states at the bottom of the 1D parabola (on the left, see Supplementary Note 1 for a discussion of their effects) and the charge line (on the right) probing at the same time many-body states well into the nonlinear regime far away from the Fermi energy. The non-zero $\Gamma_I$ is largely caused by monolayer fluctuations in the barrier thickness, which give a spread of subband energies of this order. In Sample A, this $\Gamma_I = 0.32 \pm 0.03$ meV is about 20% larger than $\Gamma = 0.25 \pm 0.03$ meV of the 2D–2D tunnelling signal (the magenta and cyan parabolae in Fig. 2b) measured without wire gates, probably because they introduce some additional disorder. We concentrate on this region because the larger $\Gamma_I$ close to the bottom of the 1D parabola obscures the effect of $K_c$ on the conductance line shape. We choose representative cuts through the data (shown as black vertical lines in Fig. 2a). The conductance is shown schematically in the lower-left inset, and its differential (without background subtraction) in Fig. 2b. The position of the peak corresponds to the 1D parabola. The densities of the two layers are determined from the crossing points (labelled $\pm k_F$) and are used to calculate $E_F$.

The non-interacting calculation (model 1, crosses) gives a sharp drop to the left of the peak, as there are no states below the parabolic dispersion. However, the full interacting calculation (based on model 2 and exemplified by the top-most fit presented in Fig. 4a) predicts exactly this, an enhancement of tunnelling there, as it allows multiple many-body excitations to be created. An example of such an excitation is marked by the green circle in Fig. 2c, composed of a hole deep below $E_F$ ($d$-band in Fig. 2d and a number of Luttinger-liquid modes around $E_F$ ($r$-band)). This predicts a power-law dependence $\alpha$ on energy away from the dispersion relation/band, where $\alpha$ is, remarkably, a function of momentum. Note that, in calculating the tunnelling conductance, one convolves the 1D and 2D spectral functions over all momenta and energies, so the variation of $\alpha$ with $k_x$ must be included and the result at any value of DC bias includes a range of $\alpha$. Model 3 dispenses with this momentum dependence (see Fig. 4c). Neither a fit with the maximum correctly normalised (dots) nor one with the tail aligned to the data (+ symbols) matches the data at all well, as they deviate immediately from the data on one or other side of the peak. This shows that the momentum dependence of $\alpha$ is required to get a good fit. This figure also shows another attempt to fit the data using the non-interacting model (× symbols), where $\Gamma$ is increased to match the tail, but this clearly gives an unacceptably slow decay to the right of the maximum.

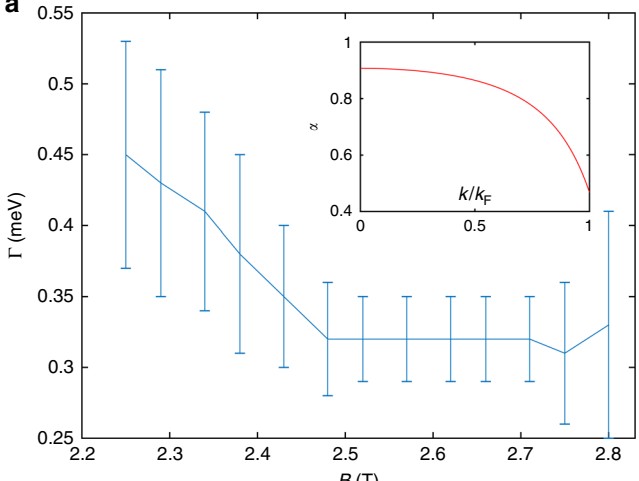

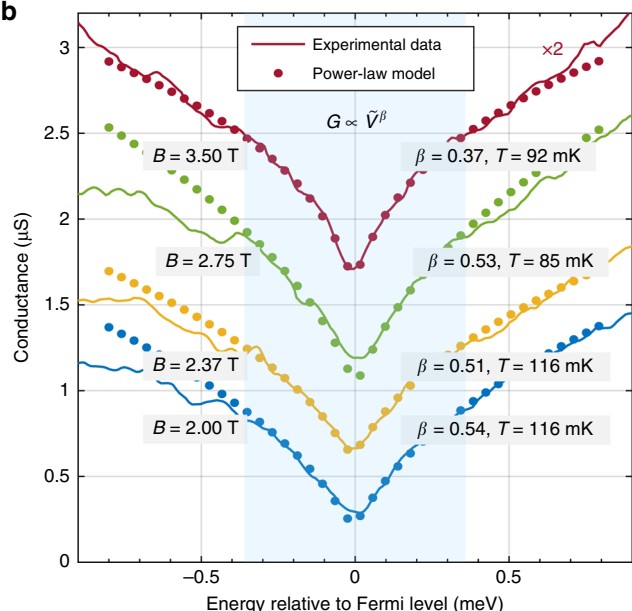

**Fig. 5** Broadening and zero-bias anomaly. **a** Dependence of the broadening fitting parameter $\Gamma$ on field $B$, for the full interacting calculation (model 2). The inset shows how $\alpha(k)$ varies with $k$, from Eq. (1). The error bars indicate the range of $\Gamma$ values for which the calculated model consistently matches the experimental data around the conductance peak, by visual comparison. **b** Power-law fit of the ZBA for the first sample. The conductance is modelled to be of the form $G = a\tilde{V}^\beta$, where $\tilde{V} = \sqrt{V_{AC}^2 + V_{DC}^2 + (3k_B T/e)^2}$. The extracted values of $\beta$ are similar to those obtained originally for carbon nanotubes[13] in the end-tunnelling regime and in our previous work[15]. A background parasitic conductance due to tunnelling in the p-region was subtracted before the power-law fit. $\beta$ is sensitive to the estimated background conductance with 50% error margin. The AC excitation voltage was $V_{AC} = 5\,\mu$V. Each sweep has been offset by 0.4 $\mu$S consecutively, for clarity

Note that we observe a good fit to the nonlinear theory in both samples studied, with different wire lengths, and at different temperatures. Supplementary Figure 2 shows the results for Sample B corresponding to those shown here for Sample A in Figs. 2 and 5. In Supplementary Note 1, we consider, and exclude, other possible causes of the enhancement of tunnelling conductance below the 1D subband edge. Supplementary Figure

3, Supplementary Table I and ref. [23] show the dependence of the tunnelling conductance and layer densities on gate voltage for the two samples.

**Other evidence for interactions.** Our samples also show the following effects that were reported in previous works: (1) power-law suppression of the tunnelling conductance around zero DC bias (zero-bias anomaly, ZBA), caused by vanishing of the tunnelling density of states of the linear TTL at the Fermi energy[27,28]. This has been seen in carbon nanotubes[13], in tunnelling between two 1D wires[14] and between a 1D wire and a 2DEG[15]. From fitting to the power-law formula used in ref. [15] in the low-energy regime, with exponent $\beta$, we deduce the finite range of energies around $EF$ (blue region in Fig. 5b), where the ZBA affects $G$ significantly (see Supplementary Note 1), so that we can exclude this region from the nonlinear or non-interacting fitting described above. From the ZBA fits, we obtain $\beta = 0.48 \pm 0.24$. This linear TLL exponent $\beta$ gives, using the relation between $\beta$ and $K_c$ in the end-tunnelling regime from ref. [15], $K_c = 0.59 \pm 0.13$ (sample A). (2) Separation of spinon and holon excitations in the linear regime[14,15] close to the Fermi momentum $\hbar k_F$ (the red lines labelled by C and S in Fig. 2a, b and e).

The intrinsic interaction parameter $K_c$ for each sample can be extracted from the slopes of the spinon and holon modes, see Fig. 2e, as $K_c = v_s/v_c = 0.76 \pm 0.07$ (sample A) and $K_c = 0.61 \pm 0.06$ (sample B). These two other ways of extracting $K_c$ are consistent with the value of $K_c = 0.72 \pm 0.03$ (sample A) and $K_c = 0.69 \pm 0.03$ (sample B) extracted in the nonlinear regime in Fig. 4a and b, giving additional confidence that we are observing an effect of the same Coulomb interaction within a 1D system in the nonlinear regime.

Another indication of electron–electron interactions is the effective mass $m_{1D}$ that we observe for the 1D parabola. For the calculated peaks to line up well in energy with the data in Fig. 4a, we find that $m_{1D} = 0.92(\pm 0.05)m^*$ (sample A) and $m_{1D} = 0.81(\pm 0.05)m^*$ (sample B), where $m^*$ is the 2D effective mass. Note that, while there is an uncertainty in the exact tunnelling distance, and hence there is in the conversion factor between $B$ and momentum, we use the 2D parabolae (measured by the 1D system or the parasitic region) to calibrate the distance. The observed difference in masses is due to non-equal contributions of the interactions in different dimensions to renormalisation of the free particle mass.

## Discussion

We have studied experimentally the decay of the tunnelling current below the bottom of the 1D subband. The conductance $G$ decays more slowly than that predicted by the non-interacting theory, or by interacting theory that includes only a fixed power law $\alpha$. A good fit, however, is obtained using a power law that depends on momentum, as predicted by the recent theory. This appears to be the first example of an interaction-driven variable power law.

## Methods

**Tunnelling current.** The tunnelling current between the two 2DEG layers is given by[29]

$$I \propto \int d\mathbf{k} dE [f_T(E - E_{F1D} - eV_{DC}) - f_T(E - E_{F2D})] \\ \times A_1(\mathbf{k}, E) A_2(\mathbf{k} + ed(\mathbf{n} \times \mathbf{B})/\hbar, E - eV_{DC}),$$

$$(2)$$

where $e$ is the electron charge, $f_T(E)$ the Fermi–Dirac distribution function, $d$ is the spatial separation between the two layers of 2DEGs, $\mathbf{n}$ is the unit normal to the surface, $\mathbf{B} = -B\hat{\mathbf{y}}$ is the magnetic-field vector (magnitude $B$), $\hat{\mathbf{y}}$ is the unit vector in the $y$-direction, $A_1$ and $A_2$ are the spectral functions of the 1D and 2D systems, respectively and their corresponding Fermi energies are $E_{F1D}$ and $E_{F2D}$. According to Eq. (2), the tunnelling current between the two layers is proportional to the overlap integral of the spectral functions of the two layers. We can induce an offset

$eV_{DC}$ in the Fermi energies between the two layers by applying a DC bias $V_{DC}$. A momentum offset can be induced by a magnetic field of strength $B$ parallel to the 2DEG layers (as shown in Fig. 3b). Assuming that the field direction is along the $y$-axis, the vector potential is equal to $\mathbf{A} = (zB, 0, 0)$ in the Landau gauge, and the Lorentz force shifts the momentum of the tunnelling electrons in the $x$-direction by $\mathbf{p} = \hbar\mathbf{k} = (-edB, 0, 0)$. At low temperatures, the Fermi–Dirac distributions can be approximated by a Heaviside step function $\theta(E)$.

**Modelling details.** The tunnelling rate is proportional to each spectral function, which gives the probability density to find an electronic state at a given point of energy–momentum space. The spectral function can be obtained via a Fourier transform of the real-space Green function[30]. The latter is expressed in terms of electron wave functions. In a free 2D space, the electron wave function is a plane wave, so the spectral function is a delta function: $A_2(\mathbf{k}, E) = \delta(E - \varepsilon(\mathbf{k}))$, where $\varepsilon(\mathbf{k})$ is the dispersion relation $\hbar^2(k_x^2 + k_y^2)/2m$. To account for disorder broadening, the spectral function is convolved with a Lorentzian function with spread $\Gamma$:

$$A_2(\mathbf{k}, E, \Gamma) = \frac{\Gamma}{\pi} \frac{1}{\Gamma^2 + \left(E - \frac{\hbar^2(k_x^2 + k_y^2)}{2m^*}\right)^2}.$$

$$(3)$$

Experimentally, the gate-induced 1D channels have finite transverse confinement potentials (instead of being infinitely narrow and hence having infinite subband spacing). For this reason, the 1D spectral function depends on the transverse dimension, $ky$. The confinement potential can be treated as a parabolic quantum well, whose electron wave function is given by a quantum harmonic-oscillator solution[31]. The confinement results in energy levels known as 1D subbands. The spectral function of the 1D system is given by the Fourier transform of the wave functions summed over all subbands, which contribute to conduction (i.e. below $E_F$), and convolved with a Lorentzian function to account for broadening. Without considering the effects of interactions, the 1D spectral function is

$$A_{1non-int}(\mathbf{k}, E, \Gamma) = \sum_n \frac{\Gamma}{\pi} \frac{H_n(k_y a) e^{-(k_y a)^2}}{\Gamma^2 + (E - E_n(k_x))^2},$$

$$(4)$$

where $n$ is the 1D subband index, $a = m_{1D}\omega/\hbar$ is a finite width of the wire in the $y$-direction, $E_n(k_x) = \hbar^2 k_x^2/(2m_{1D}) + \hbar\omega(n + 1/2)$ is the parabolic dispersion of each subband, $\hbar\omega$ is the energy spacing of the subbands and $H_n(x)$ are the Hermite polynomials. As was shown in the main text, the 1D spectral function derived from the mobile-impurity model is $A_1(k_x, E) \propto 1/|E - \varepsilon(k_x)|^{\alpha(k_x)}$ (see details in Supplementary Note 2) and the momentum dependence of the exponent is given by Eq. (1). The 1D spectral function that includes the effects of interactions is therefore

$$A_{1int}(\mathbf{k}, E, \Gamma) = \int_{-\infty}^{\infty} dz \sum_n \frac{\theta(E - E_n(k_x) - z)}{(E - E_n(k_x) - z)^{\alpha(k_x)}} \cdot H_n(k_y a) e^{-(k_y a)^2} \frac{\Gamma}{\pi} \frac{1}{\Gamma^2 + z^2},$$

$$(5)$$

where a finite number of 1D subbands $n$ is taken into account.

The integral in Eq. (2) was evaluated numerically using Mathematica, which gave the tunnelling current across the sample and was used to calculate the conductance after taking the derivative with respect to $eV$. The calculation and the experimental results were normalised to their own conduction peak values and compared in Fig. 4.

## Data availability

Data associated with this work are available at the University of Cambridge data repository (https://doi.org/10.17863/CAM.39711).

## Code availability

Code associated with this work is available as part of the data available at the above link.

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

## Acknowledgements

This work was supported by the UK EPSRC [Grant Nos. EP/J01690X/1 and EP/J016888/1]. O.T. acknowledges support from the German DFG through the SFB/TRR 49 programme. L.G. acknowledges support from NSF DMR Grant No. 1603243.

## Author contributions

Project planning: C.J.B.F., O.T., L.I.G. and A.J.S.; MBE growth: I.F. and D.A.R.; e-beam lithography: J.P.G.; sample fabrication: Y.J. and M.M.; transport measurements: Y.J., M.M., A.A., W.K.T. and C.J.B.F.; analysis of results and theoretical interpretation: Y.J., C.J.B.F. and O.T.

## Additional information

**Competing interests:** The authors declare no competing interests.

