## [Peer Review File · Nature Communications]

Reviewers' comments:

Reviewer #1 (Remarks to the Author):

This paper describes the energy-momentum dependence of tunneling current between 1D and 2D electron systems, where non-linear Luttinger liquid can be studied. While the subject is interesting, I do not think the main statement 'Momentum-dependent power law' is 'measured' in this work. I do not recommend this paper for publication.

The authors claim that 'momentum dependence of the power law in the high-energy region' is observed. I am afraid this is not clearly demonstrated.

Firstly, the main experimental data in Fig. 3 shows an almost symmetric peak due to the disorder (Γ). Many body states should enhance the low (more negative) energy tail in the conductance profile, but this is seen neither in the experimental data nor in model 2 (nonlinear TLL). It is not clear to me why the calculated plots for the non-interacting case show an asymmetric peak with enhancement in the high (less negative) energy tail in the forbidden region.

Secondly, 'momentum dependence of the power law' is not shown. Based on the exponent shown in the inset to Fig. 4(a) and the conditions [black lines in Fig. 1(d)], the exponent α is almost constant within the investigated range. This could be the reason why no significant B dependence (other than the peak shift) is seen in the conductance profile of Fig. 3.

Thirdly, I do not see any power law dependency in the experimental data. The authors used a model with a momentum dependent power law, but this is not experimentally investigated.

Minor and/or optional comments

- The 1D quantization energy should be described.
- Is Figure 1(a)(iii) correct? 1D parabolic bands should be drawn along k_x axis (not k_y axis). The two green lines do not look to be the overlap of the 1D and 2D bands.
- Awkward coloring in Fig. 1(d). The authors should avoid color (yellow and blue) shading on a color plot in Fig. 1(d). The same colors used as the background of the insets are misleading.
- It is hard to understand the lower-left inset to Fig. 1(d). The conditions of the three lines (ribbons?) are not shown. Did the authors see the power law dependences?
- Line 33: The authors should concisely explain the mobile impurity model in the main text.
- Line 87: I did not understand what is 'Replica of the principal 1D dispersion just above kF '.
- Line 90: It is unable to understand what 'parasitic tunneling' is, before reading Methods. Maybe one can move this together with the subtraction scheme to Methods.
- Line 105: $A_1(k_x, E)$ is measured directly in our experiment. Actually, the current is given by Eq. (2). Is the current (more or less) proportional to $A_1(k_x, E)$?
- Line 119: ZBA is analyzed with parameter β . I did not see any message to the non-linear Luttinger model. Is there any relation between β and α ?
- Line 127: Do you mean 'avoid detecting the localized states ...'?
- Line 128: While I do not know how the charge line influences the measurement, I am wondering why

analysis at higher B (with a large change in α shown in the inset to Fig. 4a) was avoided in this work.

- Line 172: How long is the distance calibrated with the 2D parabola? Is it close to 32 nm?

- Line 246: (3) \rightarrow (2) ?

Reviewer #2 (Remarks to the Author):
see attachment

Reviewer #3 (Remarks to the Author):

This paper examines the tunneling between 1D quantum wires and a two dimensional systems, with a control on both the energy and the momentum of the tunneling particle (provided by a gate voltage and a magnetic field). The tunneling conductance is compared with calculations of a 1D modified Tomonaga-Luttinger liquid theory, taking properly into account the effects of band curvature which were shown to lead to a modified exponent compared to the TLL one. The agreement between the observations and the theoretical calculations is claimed to be the first experimental validation of this theory.

Tomonaga-Luttinger liquids have been a paradigmatic description of one dimensional interacting quantum systems, with predictions that have been extremely relevant and successfully tested for a variety of quantum systems both in condensed matter and cold atomic gases.

Understanding the deviations from this theory is thus an important challenge of the field. Recently a beautiful theory was developed

(see Refs 16,18 and 25 of the present paper) properly incorporating the effects of band curvature, and computing its consequences for properties close to a threshold (thus at finite energy, which is why the band curvature is not irrelevant). In particular this led to a different, and momentum dependent, exponent than the one predicted by TLL. In that respect the effect is quite similar to the modification of the threshold exponent by interactions with a localized impurity in the celebrated X-ray edge problem. Testing experimentally such a theory is thus an important step, and indeed a very relevant research line. I thus think that the research presented in the present paper is worthy of publication in Nature communications if the claim of the paper is met.

I have however several questions concerning this point on which I would like the answers from the authors before I can make a positive recommendation:

- I have no reason to doubt the experimental results which are done in system which is well suited in principle to test the theoretical predictions (this is a systems similar to the one used by e.g. the Yacoby group to probe the spin-charge separation). I would however like to have more details given in the paper on the screening of the Coulomb interaction in such a system. Indeed the standard TLL predictions (and the models with constant exponents) are valid for an interaction for which the q dependence can be neglected. For coulomb interaction the TLL parameters would acquire a momentum dependence by themselves (i.e. independently of the effect of mobile impurity). Can one safely ignore this effect here at the energies at which the author want to do the experiment-theory comparison ? More details could be given on that point.

- Concerning the comparison between theory and experiment (Fig.3) which is the heart of the present paper I am confused by the region of energy in which the authors make the comparison. The theory of the mobile impurity is in principle a theory which is built on the top of the TLL theory by adding the effect of deep particle-hole excitations. It indeed leads to a modification of the exponent compared to TLL but for the theory to be valid one should already be in a regime of energy for which the TLL theory should have been a good approximation of the physics of the system, i.e for energies not too far from the zero of energy for the excitations, and in particular small compared to the bandwidth of the material. Then in this regime there is a second (and lower) energy scale at which the effects of curvature manifest themselves leading to a modification of the exponent.

I would thus have naively expected that the test of the theory should be made close to the zero of energy in Fig.3. I am ready to buy that for some reason one has to correct for another effect leading to a zero bias anomaly (in principle the ZBA could also be due to the exponents of TLL or modified TLL, so perhaps one more sentence on the "non conservation of momentum" mentioned in the paper could also be useful). However given the fact that the bandwidth is about 3 mV (according to Fig.1) I would have thought that the test of the powerlaw should be made between zero and say 1 or 1.5 mV to be safely in the regime where we can use the field theory (discussed in the SM). However in this regime the Fig.3 shows excellent agreement between noninteracting and interacting theory (why ?) in (a). This is surprising if K_{ρ} is taken smaller than one. It also shows good agreement with models with constant exponents in panel (c) (showing here clear deviations with the non interacting theory).

So first I am not sure to understand the difference between the noninteracting models in the two panels. Is this entirely due to the phenomenological broadening ? Second I would say that in the low energy regime (where we would expect TLL or the modified TLL to be valid) there is satisfactory agreement with a model with constant exponent (even better for e.g. the lower curve than with the interacting model of panel (a)).

But the authors claim that the comparison should be made around the peak of conductance, which is at an energy of about -2 mV. So first I have a priori a severe problem with using TLL or a theory built on the top of it so deep in the band since there are many irrelevant operators that will affect it at that energy (spin-charge coupling, higher harmonics etc.). Second if I look at the data around the peak the model with a constant exponent α_2 in panel (c) provides a decent fit of the theory except at energies even lower than the peak in which it is quite natural to expect the TLL theory to severely degrade. If the modified theory is good at high energy why is it failing at low energies (see e.g. the lowest curves of panel (a))

So with the level of information given the paper it is difficult to really ascertain how successful is the present experiment in testing precisely the modified TLL theory and the new value of the exponents it predicts.

The authors should provide more details on the above mentioned points since this is of course the heart of the present paper.

Jin et al – Momentum-dependent power law measured in an interacting quantum wire beyond the Luttinger limit.

In this work, the authors have convinced me that most likely they have observed an interaction-driven variable power law even in the presence of a length scale.

More specifically, they use a clever 1D-2D transport-spectroscopy technique to present what is perhaps the first experimental validation of the power law momentum dependence of spectral function for interacting electrons in the high-energy region, as predicted in the nonlinear-Luttinger-liquid theory for a one-dimensional quantum fluid (mobile-impurity model).

The experiment is clearly complex, and while I consider myself able to understand it all, if I have used “most likely” and “perhaps” above it is because the paper is extremely hard to read. While I can see there are so many parameters they could have opted to vary, they have done due diligence by probing different devices with different 1D lengths, and by observing what was previously known about LL and LL beyond the linear approximation. This being said, my main criticism is that **even if know their previous works** I frankly needed to make a leap of faith in agreeing with their conclusions. My second most important criticism is that the paper hinges on modelling, and the data is not stand-alone *per se*. There is nothing wrong with this, but again some “leap of faith” had to be done given that the effect is not dramatic. But it seems to be there.

The paper has also left me with so many questions unanswered. Effects of subband squeezing due to magnetic field. Effects of disorder due to averaging, lithography roughness, potential aluminum content fluctuation, and density. How do they know that all air bridges worked? I presume Fig. 4 is all done with a single subband, but was the same true in the carbon nanotubes experiment for which beta parameter is found to be conspicuously similar? On top of that, many other parameters in NTs could play a role, such as density for instance. Is it a coincidence? The authors do not have to answer these questions, but I am sure the community will want answers to these at some point. Thus, the authors should not take negatively any of the comments I mentioned above.

My verdict is the following:

Is the manuscript, *per* the Nature Communications standards, a paradigm shift and a serious advance. My answer is YES, with the reservations that I don't see a smoking gun for non-linear LL but rather a well-executed and sound experiment for which the most “honest” interpretation point towards the non-linear mobile impurity model.

Is the manuscript clear and can be read by a non-specialist? My answer is a clear NO. I think even for PRL it would be a hard sell. The problem is that it is an extremely complex experiment and any LL theories are equally and notoriously difficult to explain in a concise way. All in all, I do recommend publication for Nature Communications but with the following veto: in order to increase impact and to help the community, the paper could, and *should* be packaged in a more pedagogical way. As an example, Fig.1 is so complex,

the cartoons, and the yellow shades are impossible to see even when I printed the manuscript in a full page, and in colour. Fig.1 C is very poor, and I would think could have shown after Fig.2. Fig.1 could easily have been split in two, and I do not understand why the paper was packaged in this way (to save space? I think Nature Communications should be flexible). There is also a constant back and forth between experimental and theoretical content that while at times does work, at others it does not.

In closing, I demand the authors to present the manuscript in a better way, and I hope they will go beyond the few examples I have mentioned above. I will gladly re-read their second version and will approve for publication once the authors make such effort.

Response to referees

Reviewer #1:

1) This paper describes the energy-momentum dependence of tunneling current between 1D and 2D electron systems, where non-linear Luttinger liquid can be studied. While the subject is interesting, I do not think the main statement 'Momentum-dependent power law' is 'measured' in this work. I do not recommend this paper for publication.

The authors claim that 'momentum dependence of the power law in the high-energy region' is observed. I am afraid this is not clearly demonstrated.

Firstly, the main experimental data in Fig. 3 shows an almost symmetric peak due to the disorder (Γ). Many body states should enhance the low (more negative) energy tail in the conductance profile, but this is seen neither in the experimental data nor in model 2 (nonlinear TLL). It is not clear to me why the calculated plots for the non-interacting case show an asymmetric peak with enhancement in the high (less negative) energy tail in the forbidden region.

We are disappointed that the referee does not think we have demonstrated the momentum dependence of the new power law clearly. The referee is correct that the low-energy tail should be enhanced, and this is exactly what we observe

and discuss in detail. We hope in our revised version of the paper that the description is much clearer so that the referee can now appreciate this better.

In both the interacting and non-interacting cases, we calculate the complete spectrum, including tunnelling from the Fermi energy of either system (1D or 2D) to the dispersion of the other system, giving a pair of parabolae. Lorentzian broadening is included to allow for monolayer fluctuations of the well and barrier widths. This, together with the convolution of the 1D and 2D spectral functions, provides some conductance in the region between the upright and upturned parabolae (the ‘forbidden region’). This broadening also accounts for part of the conductance observed in the more-negative energy tail, but it is *not* correct to say that the many-body enhancement of the conductance is not observed there as well, experimentally or in the nonlinear TLL model—that is exactly what we show in the original Fig. 3(a) and (b). This is where the interacting model differs from the non-interacting one. It is only the asymmetry provided by the interacting model with a momentum-dependent exponent that allows our data to be fitted there.

2) Secondly, ‘momentum dependence of the power law’ is not shown. Based on the exponent shown in the inset to Fig. 4(a) and the conditions [black lines in Fig. 1(d)], the exponent α is almost constant within the investigated range. This could be the reason why no significant B dependence (other than the peak shift) is seen in the conductance profile of Fig. 3.

The many-body enhancement to the left of the peak discussed above *does* change significantly with B . In the original Fig. 3(c) (now 4(c)) we show *two* attempts to fit our data using the nonlinear TLL model with artificially constant α . One can fit the left side of the peak only with a very large value of γ (making α irrelevant, but giving an impossibly large conductance on the right of the peak), or else fit the right side of the peak with a smaller value of γ and then see too little conductance on the left, whatever value of α we choose. So the referee’s claim that there is no evidence for the momentum dependence of the power law α is wrong.

3) Thirdly, I do not see any power law dependency in the experimental data. The authors used a model with a momentum dependent power law, but this is not experimentally investigated.

The power-law dependency shows up as the slower-than-expected drop-off in the conductance to the left of the peaks in the original Fig. 3 (now 4), as discussed above. The Lorentzian broadening drops off considerably faster than the data, leaving the power-law as the only plausible explanation.

4) Minor and/or optional comments

- The 1D quantization energy should be described.

The 1D subband spacing is a few meV, as can be found from the spacing between the peaks of the first and second 1D subbands at the bottom of their parabolae. We have now mentioned this in the text.

5) - Is Figure 1(a)(iii) correct? 1D parabolic bands should be drawn along k_x axis (not k_y axis). The two green lines do not look to be the overlap of the 1D and 2D bands.

We thank the referee for pointing out that the axes used in Figure 1(a)(iii) do not match the axes shown in Figure 1(a)(i). (iii) is rotated to show the overlap as well as possible. The green lines are just guides to the eye, to indicate where the Fermi surface of the 2D system (blue paraboloid) intersects the more complex 1D spectral function. They have been adjusted to be more accurate now that the plot is larger (in the new Fig. 1(c)).

6) - Awkward coloring in Fig. 1(d). The authors should avoid color (yellow and blue) shading on a color plot in Fig. 1(d). The same colors used as the background of the insets are misleading.

We thank the referee for pointing this out and have changed the visual representation of these features in the original Fig. 1(d) (now 2(a)). We have retained dark yellow for the colour of the dashed line of the box surrounding the region, in attempt still to link the yellow regions in each figure.

7) - It is hard to understand the lower-left inset to Fig. 1(d). The conditions of the three lines (ribbons?) are not shown. Did the authors see the power law dependences?

These curves are just schematic representations of how G might decay with energy, differently for each momentum k .

8) - Line 33: The authors should concisely explain the mobile impurity model in the main text.

The mobile-impurity model is already introduced and its position on the general landscape of the theoretical physics is already explained, along with the references to the original papers and to the main review of this field, on the first page of the manuscript. A theoretical construction of this model and the explicit calculation of its predictions in the particular case of spinful fermions with repulsive interaction used for fitting the results of our experiment in the main text are summarised on more than 2 pages of section 2 of the Supplementary information, where it can be easily found by a interested reader. We feel strongly that it should not be included in the main text for two reasons: even in a brief form it is too long for a Nature Communication paper and a lengthy theoretical discussion does not reflect the experimental nature of our work.

9) - Line 87: I did not understand what is 'Replica of the principal 1D dispersion just above kF '.

We have improved the description of replicas—formerly we had referred the reader to our previous papers which covered them in detail.

10) - Line 90: It is unable to understand what 'parasitic tunneling' is, before reading Methods. Maybe one can move this together with the subtraction scheme to Methods.

As this is a significant correction to the data, we do not wish to hide it in Methods. We have moved all of the discussion of it from the Methods to the main text, to the part where we describe the experiment. We have also improved this description to make it as clear as possible for the reader.

11) - Line 105: 'A1(kx , E) is measured directly in our experiment'. Actually, the current is given by Eq. (2). Is the current (more or less) proportional to A1(kx , E)?

The current is a convolution of A1(k , E) and A2(k , E) over momentum and energy, see Methods. We thank the referee for this remark and have corrected the language in the original Line 105 (now 277) to remove the word 'directly'. The current is roughly proportional to A1(kx , E) in the region of interest, but fitting a power law to the conductance is certainly not adequate.

12) - Line 119: ZBA is analyzed with parameter beta. I did not see any message to the non-linear Luttinger model. Is there any relation between beta and alfa?

The message from the ZBA is the region of the linear TLL, see the blue region in the original Figs. 3 and 4 (now 4 and 5) and the corresponding description in the main text. The nonlinear Luttinger liquid is valid outside this blue region. In the final section of the paper we now mention the values of K_c we derive from β (near line 399). The value is compatible with our values from the nonlinear model fits, within the error.

13) - Line 127: Do you mean 'avoid detecting the localized states ...'?

No, we are listing three items, so have added "as well as" after "avoid the 2D parabola" to clarify this (new line 334).

14) - Line 128: While I do not know how the charge line influences the measurement, I am wondering why analysis at higher B (with a large change in α shown in the inset to Fig. 4a) was avoided in this work.

For higher B close to the Fermi point the interacting 1D system is in the linear regime of the usual TLL, which has already been measured in 1D-1D tunnelling [Science 295, 825 (2002)] and by us [Science 325, 597 (2009)]. In order to study the nonlinear Luttinger liquid in this work we have stayed at a sufficient distance from it at smaller B . The charge line starts to enhance the conductance at the left edge of the top curve in the new Fig. 4(b), which is why we exclude that part from the fits.

15) - Line 172: How long is the distance calibrated with the 2D parabola? Is it close to 32 nm?

We use a value of 35 nm (this value was chosen because it produced the best overall parabola curvature that matches the density plot of the conductance in the original Fig. 1 (now 2)), very close to the nominal 32 QW centre-to-centre distance. It is reasonable that the wave functions are not centred in the wells, as they are not symmetric because there is a bias between the wells. In addition, some miscalibration in the MBE growth is possible. To make both the 1D and 2D parabolae line up with the data, we have to introduce a renormalised 1D mass (see the new line 415).

16) - Line 246: (3) \rightarrow (2) ?

Yes, thanks for pointing this out.

Reviewer #2:

17) In this work, the authors have convinced me that most likely they have observed an interaction-driven variable power law even in the presence of a length scale. More specifically, they use a clever 1D-2D transport-spectroscopy technique to present what is perhaps the first experimental validation of the power law momentum dependence of spectral function for interacting electrons in the high-energy region, as predicted in the nonlinear-Luttinger-liquid theory for a one-dimensional quantum fluid (mobile-impurity model).

We are pleased that the referee is convinced by our data.

18) The experiment is clearly complex, and while I consider myself able to understand it all, if I have used ‘most likely’ and ‘perhaps’ above it is because the paper is extremely hard to read. While I can see there are so many parameters they could have opted to vary, they have done due diligence by probing different devices with different 1D lengths, and by observing what was previously known about LL and LL beyond the linear approximation.

We have now tried to make the paper much easier to read by rearranging and expanding various parts, at the expense of lengthening it considerably.

19) This being said, my main criticism is that even if know their previous works I frankly needed to make a leap of faith in agreeing with their conclusions. My second most important criticism is that the paper hinges on modelling, and the data is not stand-alone per se. There is nothing wrong with this, but again some ‘leap of faith’ had to be done given that the effect is not dramatic. But it seems to be there.

We have endeavoured to give all the experimental and theoretical details so that the reader can see that we have done as good a job as possible of making the two comparable. The transport-spectroscopy technique unavoidably involves a convolution, but the functions are sharply peaked and so the result resembles the individual spectral functions. Hence we observe the various parabola. Interpretation of the line-shape of each peak is more elaborate, and so we have to rely on a calculation to ensure that we have taken into account all contributions to the tunnelling at each value of B and V_{DC} . The spectral function of a 2DEG and of a non-interacting 1D electronic system (which we use as a reference point) are well established. Thus, the only new object in the calculation is the 1D spectral function with interactions, which accounts for the additional contribution expected from the power law. It is clear from the original Figs. 3 (a), (b) and (c) (now Fig. 4) that the non-interacting model cannot fit the full line-shape of the 1D system, which is very clearly asymmetric beyond the disorder broadening, and that it can only be explained using a 1D model with interactions.

20) The paper has also left me with so many questions unanswered. Effects of subband squeezing due to magnetic field. Effects of disorder due to averaging, lithography roughness, potential aluminum content fluctuation, and density. How do they know that all air bridges worked? I presume Fig. 4 is all done with a single subband, but was the same true in the carbon nanotubes experiment for which beta parameter is found to be conspicuously similar? On top of that, many other parameters in NTs could play a role, such as density for instance. Is it a coincidence? The authors do not have to answer these questions, but I am sure the community will want answers to these at some point. Thus, the authors should not take negatively any of the comments I mentioned above.

‘Subband squeezing’: there should be little effect of the magnetic field on the wave functions as the field is perpendicular to the 1D confinement. There is already strong confinement vertically by the quantum well, so at these fields the well confinement dominates over any magnetic confinement.

‘Effects of disorder due to averaging, lithography roughness, potential aluminium content fluctuation’: measuring the current from all the wires in parallel averages over many different impurity configurations and wire-width fluctuations. We have previously shown that doubling the length of the wires in such an

array doubles the current whilst leaving the conductance structure unchanged, at least when the wires are slightly less pinched off. So there is evidence that most wires contribute to tunnelling, rather than a few weak points (which would not conserve momentum).

The air-bridge was checked post-fabrication with an optical microscope (We avoided using an SEM to avoid damaging the device by accidentally implanting impurities into the sample). While it is impossible to distinguish individual air bridges or finger gates, the entire 1D-array region has a highly uniform, and iridescent, appearance. In our experience, any serious fabrication fault will disturb the uniform appearance and show up as a distinctive local anomaly under the microscope.

Due to the thickness of metal applied (over 100 nm), we have never observed an air-bridge breaking. The most-regularly observed failure is of an entire strand of air-bridge becoming detached from the base finger-gate array. In this case, the loose air-bridge will redeposit on to a nearby top gate structure, creating a short between different gates. We can easily check for such faults by testing for continuity between the finger-gate array and the other gates. Having observed many faulty samples, we are confident that the data presented came from devices free of faults in the air-bridge array.

In addition, we use two sets of air bridges connecting the finger gates together, so that if there is a poor contact at one end of a finger, there is a good chance that the connection is good at the other end.

Electrically, we would observe clear signatures of regions without complete finger gates. They would look like a 2D parasitic region, but should be strongly affected by the surrounding finger gates. We do not see signs of such a strong dependence on finger-gate voltage in the parasitic parabolae.

‘I presume Fig. 4 is all done with a single subband, but was the same true in the carbon nanotubes experiment for which beta parameter is found to be conspicuously similar? On top of that, many other parameters in NTs could play a role, such as density for instance. Is it a coincidence?’:

It is hard to say why K_c should be similar in both our wires and CNTs, but it has to lie between 0.5 and 1 anyway. CNTs may provide stronger confinement (and hence interactions) but this may be offset by screening from multiple subbands.

21) My verdict is the following:

Is the manuscript, per the Nature Communications standards, a paradigm shift and a serious advance. My answer is YES, with the reservations that I don't see a smoking gun for non-linear LL but rather a well-executed and sound experiment for which the most 'honest' interpretation point towards the non-linear mobile impurity model.

Is the manuscript clear and can be read by a non-specialist? My answer is a clear NO. I think even for PRL it would be a hard sell. The problem is that it is an extremely complex experiment and any LL theories are equally and notoriously difficult to explain in a concise way. All in all, I do recommend publication for Nature Communications but with the following veto: in order to increase

impact and to help the community, the paper could, and should be packaged in a more pedagogical way. As an example, Fig.1 is so complex, the cartoons, and the yellow shades are impossible to see even when I printed the manuscript in a full page, and in colour. Fig.1 C is very poor, and I would think could have shown after Fig.2. Fig.1 could easily have been split in two, and I do not understand why the paper was packaged in this way (to save space? I think Nature Communications should be flexible). There is also a constant back and forth between experimental and theoretical content that while at times does work, at others it does not.

In closing, I demand the authors to present the manuscript in a better way, and I hope they will go beyond the few examples I have mentioned above. I will gladly re-read their second version and will approve for publication once the authors make such effort.

We thank the referee for these helpful comments. We have done our best now to make the paper much more readable and pedagogical, and hope the referee will find it understandable now.

Reviewer #3: (OT)

22) This paper examines the tunneling between 1D quantum wires and a two dimensional systems, with a control on both the energy and the momentum of the tunneling particle (provided by a gate voltage and a magnetic field). The tunneling conductance is compared with calculations of a 1D modified Tomonaga-Luttinger liquid theory, taking properly into account the effects of band curvature which were shown to lead to a modified exponent compared to the TLL one. The agreement between the observations and the theoretical calculations is claimed to be the first experimental validation of this theory.

Tomonaga-Luttinger liquids have been a paradigmatic description of one dimensional interacting quantum systems, with predictions that have been extremely relevant and successfully tested for a variety of quantum systems both in condensed matter and cold atomic gases. Understanding the deviations from this theory is thus an important challenge of the field. Recently a beautiful theory was developed (see Refs 16,18 and 25 of the present paper) properly incorporating the effects of band curvature, and computing its consequences for properties close to a threshold (thus at finite energy, which is why the band curvature is not irrelevant). In particular this led to a different, and momentum dependent, exponent than the one predicted by TLL. In that respect the effect is quite similar to the modification of the threshold exponent by interactions with a localized impurity in the celebrated X-ray edge problem. Testing experimentally such a theory is thus an important step, and indeed a very relevant research line. I thus think that the research presented in the present paper is

worthy of publication in Nature communications if the claim of the paper is met.

We are pleased that the referee finds this experimental work an important test of the recent theory, and worthy of publication in Nature Communications.

23) I have however several questions concerning this point on which I would like the answers from the authors before I can make a positive recommendation:

- I have no reason to doubt the experimental results which are done in system which is well suited in principle to test the theoretical predictions (this is a systems similar to the one used by e.g. the Yacoby group to probe the spin-charge separation). I would however like to have more details given in the paper on the screening of the Coulomb interaction in such a system.

While a direct measurement of the two-body interaction potential is not accessible in this experiment, it is expected to have a screened Coulomb shape, as is usually the case in semiconductors. The screening for electrons within the lowest subband of a wire comes from the 2D electron gas in the well below it and from any higher 1D subbands within the same wire. By controlling voltages on different gates (see original Fig. 2(a), now 3(a)), we can change both the 2D density and occupation of the second 1D subband. In one of our previous experiments on semiconductor wires of the same design [PRB 93, 075147 (2016)], we have found that the latter alters the v_s/v_c ratio by about 20%. This change of the interaction energy in the system can be attributed only to the change of screening radius since other parameters of the Coulomb interaction are highly unlikely to be changed by these relatively small changes of gate voltages.

We thank the referee for asking this question and have performed the same analysis as was done in [PRB 93, 075147 (2016)] in this work, demonstrating that v_s/v_c , and thus the interaction radius, for sample A can be altered by 10–30% by changing the finger-gate voltage. This new data has been added to the Supplementary Information as Fig. 7 and Table. 1.

24) Indeed the standard TLL predictions (and the models with constant exponents) are valid for an interaction for which the q dependence can be neglected. For coulomb interaction the TLL parameters would acquire a momentum dependence by themselves (i.e. independently of the effect of mobile impurity). Can one safely ignore this effect here at the energies at which the author want to do the experiment-theory comparison ? More details could be given on that point.

Undeniably, the linear TLL model is valid only close to the Fermi energy. Away from it the universal model becomes altered by, for example, details of the interaction potential [PRB 47, 16205 (1993), PRB 60, 4571 (1999)] or by $1/m$ -corrections due to a finite parabolicity of the original single-particle spectrum [JPhys C 14, 2585 (1981), JPhys Cond. Matt. 10 , L533 (1998)].

In this experiment we test the validity of the standard TLL model by fitting its predictions for conductivity. A detailed study was performed in one of our previous experiments in [Science 325, 597 (2009)] where a power law with a constant exponent was measured in conductance as a function of voltage and temperature close to the Fermi energy for up to 2 decades. In this experiment we assess the validity of the linear TLL by fitting a constant power law for higher and higher biases away from E_F in the original Fig. 4(b) (now 5(b)). We observe no visible deviations for up to $V_{dc} \simeq 0.4$ meV away from E_F , showing that any corrections to the standard TLL can be neglected in this finite energy range in this experiment.

We thank the referee for pointing out this issue and have added a corresponding discussion to the text together with some additional references.

25) - Concerning the comparison between theory and experiment (Fig.3) which is the heart of the present paper I am a confused by the region of energy in which the authors make the comparison. The theory of the mobile impurity is in principle a theory which is built on the top of the TLL theory by adding the effect of deep particle-hole excitations. It indeed leads to a modification of the exponent compared to TLL but for the theory to be valid one should already be in a regime of energy for which the TLL theory should have been a good approximation of the physics of the system, i.e for energies not too far from the zero of energy for the excitations, and in particular small compared to the bandwidth of the material. Then in this regime there is a second (and lower) energy scale at which the effects of curvature manifest themselves leading to a modification of the exponent.

We respectfully point out that the mobile-impurity theory is actually valid only at energies sufficiently far away from the Fermi point. The impurity itself describes a hole deep under the Fermi energy left by a particle that is removed from the system as a part of creating a many-body excitation at high energy. For the lowest possible excitation at a fixed momentum this corresponds to creating only a hole exactly on the spectral edge of a 1D system at any energy, in the linear and the nonlinear regimes. The mobile-impurity theory (nonlinear Luttinger liquid, nLL) originally proposed in [PRL 96, 195405 (2006), PRL 102, 126405 (2009)] combines such a deep hole with a number of modes of Luttinger liquid in order to describe excitations in the proximity of the spectral edge, at the energies at which nonlinearity of the single-particle dispersion is already essential.

The mobile-impurity part within this nLL model is well-defined only sufficiently far away from the Fermi energy. Close to the Fermi energy the modes of the TLL part of nLL can create many holes at the spectral edge, making the whole nLL with only one single hole not well-defined. The nLL model has another limitation in its region of validity (in the nonlinear regime). Since modelling of excitations further and further away from the spectral edge (see the original Fig. 6, now S1) requires inclusion of higher and higher energy modes of TLL, the nLL can only describe the many-body states within a finite energy

band around the spectral edge at high energy, the extent of which is limited by the finite energy region of validity of the TLL model from low energies.

The construction of the mobile-impurity model is already described in detail in the manuscript in section 2 of the Supplementary Information, including a comment on its validity in a finite energy band around the spectral edge, a visual representation of its construction in the original Fig. 6 (now S1), and a reference to a good in-depth review on this topic [RMP 84, 1253 (2012)]. We thank the referee for asking this question and have added explicit remarks about validity of the nLL only away from the Fermi energy in the text.

26) I would thus have naively expected that the test of the theory should be made close to the zero of energy in Fig.3. I am ready to buy that for some reason one has to correct for another effect leading to a zero bias anomaly (in principle the ZBA could also be due to the exponents of TLL or modified TLL, so perhaps one more sentence on the "non conservation of momentum" mentioned in the paper could also be useful).

No leap of faith is required to understand ZBA in this experiment. It is a well established effect of the linear TLL at low energy, see also a reply to a previous question above. The tunneling density of states of the TLL vanishes around E_F as a power-law that causes the conductance to vanish as well as a function of temperature or bias with corresponding exponents, which were evaluated theoretically in [PRL 79 5086 (1997), PRL 79, 5082 (1997)] and were measured experimentally in [Nature 397, 598 (1999), Science 295, 825 (2002), Science 325, 597 (2009)]. We find nothing new in addition to this well-known result in the conductance measured in this experiment at low energy around E_F in the original Fig. 1(d).

We agree with the referee that a better explanation of this point is needed and have added a corresponding sentence with a few extra references in the main text.

27) However given the fact that the bandwidth is about 3 mV (according to Fig.1) I would have thought that the test of the powerlaw should be made between zero and say 1 or 1.5 mV to be safely in the regime where we can use the field theory (discussed in the SM).

We respectfully disagree with this assertion. The nLL (observation of its prediction is the main point of our work) is valid far away from the Fermi energy, see a more detail reply to a previous question above and as already explained in the Supplementary Information. We should also point out here that the momentum-dependent power law in the spectral function of the nLL exists only above an appreciable threshold in energy, at quite large energies away from the Fermi energy. For the energies below the spectral threshold (thick black line in the hole sector in the original Fig. 1(d) and (e), now 2(a) and (b)) the density of states of 1D system is zero. Above this threshold the density of states is finite and this is where the nLL predicts a new, momentum-dependent power law. This is already described in the manuscript. For the cuts in the original Fig. 3

(now 4) it means that the power-law of the nLL is at energies corresponding to negative biases below about 2 to 2.5 mV (see at which energies they cross the spectral threshold in the original Fig. 1(d), now 2(a)). Please see also the next reply about the transport theory in the nonlinear regime.

28) However in this regime the Fig.3 shows excellent agreement between noninteracting and interacting theory (why ?) in (a). This is surprising if K_ρ is taken smaller than one. It also shows good agreement with models with constant exponents in panel (c) (showing here clear deviations with the non interacting theory). So first I am not sure to understand the difference between the noninteracting models in the two panels. Is this entirely due to the phenomenological broadening ?

Actually, the non-interacting theory matches neither the interacting theory nor the experimental data in the original Fig. 3 (now 4). The conductance that is measured in this experiment is calculated as a convolution of the 2DEG spectral function with different model spectral functions of the 1D system. Since the 2DEG spectral function projected on the direction of the wire (by such a convolution) is not delta-function sharp, the comparison with the measured conductance has to be made over a large enough band of energy, naturally where the models still have to be applicable. In the original Fig. 3 (now 4) the non-interacting model cannot fit the data: it either misses spectral weight at more negative biases below the 1D parabola (presented in the original Fig. 3(a), now 4(a)) or, if the broadening within the non-interacting model is increased, does not fit the data for less-negative biases above the 1D parabola (as presented in the original Fig. 3(c), now 4(c)). Thus, we conclude that a non-interacting theory, in which there is only one free parameter (broadening), cannot describe the result of our experiment. On the other hand the nonlinear interacting theory with a momentum-dependent exponent fits the observed conductance for the whole range of energies, except the ZBA at low energy (described by the linear TTL) and the second 1D subband that occurs at smaller 1D momenta.

We thank the referee for asking this question. The calculation of conductance as a convolution of 2D and different 1D spectral function is already described in detail in the Methods Section. We have put extra emphasis on comparing the conductance in a wide range of energies at the point where we describe the fits with the nonlinear models and have added more detail about validity of the linear and the nonlinear interaction theories.

29) Second i would say that in the low energy regime (where we would expect TLL or the modified TLL to be valid) there is satisfactory agreement with a model with constant exponent (even better for e.g. the lower curve than with the interacting model of panel (a)).

The discrepancy between the model with constant α and the measured data is even worse than between the non-interacting model. We present the fitting with the constant α model to demonstrate that just adding a power-law function

with an arbitrary constant exponent in the nonlinear regime does explain the result of our experiment. Using the exponent and a phenomenological broadening as two free fitting parameters can explain conductance at energies just above or just below the 1D parabola (as presented in the original Fig. 3(b), now 4(b)), but not in the whole energy range in the nonlinear regime accessible in our experiment. Note that since the conductance is a convolution of two spectral functions, fitting in a wide energy window is required, see also the previous reply for more details about this.

30) But the authors claim that the comparison should be made around the peak of conductance, which is at an energy of about -2 mV. So first I have a priori a severe problem with using TLL or a theory built on the top of it so deep in the band since there are many irrelevant operators that will affect it at that energy (spin-charge coupling, higher harmonics etc.).

We indeed do the measurement mainly in the nonlinear regime, far away from the Fermi points where the linear TLL is not valid. However, the nLL is valid exactly in this regime being the relevant field theory that describes the many-body physics at high energy, see details in previous replies above. It describes coupling between a heavy hole on the spectral edge of nonlinear spin excitations with charge excitations. It is described already in detail in the Supplementary Information, see Eq. (7) and text around it. The second 1D subband (harmonic) is far away from the main, first subband, at least 2meV away and is visible only closer to the zero 1D momentum for Sample A while the second subband is almost completely depleted for Sample B. We thank the referee for raising this issue and have added an extra remark about the second subband in the text.

31) Second if I look at the data around the peak the model with a constant exponent α_2 in panel (c) provides a decent fit of the theory except or energies even lower than the peak in which it is quite natural to expect the TLL theory to severely degrade. If the modified theory is good at high energy why is it failing at low energies (see e.g. the lowest curves of panel (a))

In fact the model with a constant α_2 does *not* fit the data in the whole range of energy (it does so only for energies lower than the peak while the discrepancy for energies higher than the peak is large) in the original Fig. 3(c) (now 4(c)) and also the nLL is valid exactly at high energy, where we compare it with our experimental data, and is not valid at low energy, where part of it—the heavy hole—becomes indistinguishable from the holes that form the density-wave modes of the TLL part and, thus, becomes not so well defined; see explanation of these points in full detail in replies above.

32) So with the level of information given the paper it is difficult to really ascertain how successful is the present experiment in

testing precisely the modified TLL theory and the new value of the exponents it predicts. The authors should provide more details on the above mentioned points since this is of course the heart of the present paper.

We hope that all the questions of all the referees have been answered completely and clearly now. We believe that the significantly revised version of the manuscript with improved structure and a lot of extra details and references leaves now no ambiguity in describing the results of our work for the reader.

Reviewers' comments:

Reviewer #1 (Remarks to the Author):

The paper was revised for describing experimental procedures more clearly and additional analysis which is not directly related to the non-linear Luttinger liquid. The main discussions on the non-linear Luttinger liquid are mostly untouched.

As I mentioned in the first report, the experiment is significantly influenced by the disorder or else to obscure the expected momentum-dependent power law. Data in Fig. 4 is the only experimental feature that the authors believe to be the signature of the nonlinear Luttinger liquid. It is hard for me to believe the presence of many-body effects from the nearly symmetric conductance peaks. For example, when the experimental trace (say the top trace $g(E)$ at $B = 2.66$ T for Sample A in Fig. 4) is folded along the peak position ($E = -2.2$ meV), one can find very tiny asymmetry $g(E = -2.2\text{meV} + \text{eps}) - g(E = -2.2\text{meV} - \text{eps}) < 0.1$ even at largest $\text{eps} = 1.6$ meV. It is even worse for other traces at lower B. When the data is compared with the plots for the non-interaction model, the deviation also remains 0 to 0.2 (at most) of the peak value even at the lowest energy condition. These asymmetry and deviation are comparable to the fluctuation of the experimental curve (about 10%) and the uncertainty in the background subtraction (about 5%). Generally, as the bias voltage is increased or decreased away from zero, more unwanted processes come in. This is not discussed in the paper and could be the reason for the asymmetry and deviation.

Similarly, the large disorder made the theoretical curves significantly broader. While the authors claim that the non-linear Luttinger model with disorder explains the experimental feature, this is not convincing as the accuracy of the calculation is not tested for this experiment. For example, a reference measurement that warrants the validity and accuracy may be needed to convince the readers.

I think that the paper is worthy for publication in some journal if the interpretation is made more speculative. However, I do not recommend this paper for publication in Nature Communications.

Reviewer #2 (Remarks to the Author):

First, I commend the authors for this well executed experiment. While I am still not clear the manuscript is "clear", I recommend it for publication in Nature Communications. There are times when we are "ahead" in research that is not easy to render our work pedagogical. But still, it is our duty.

As a competitor in the field, I commend the authors for having undertaken this experiment. As I said earlier, there appears to be an effect but not a smoking gun. I also commend the authors for their due diligence regarding my criticisms, and those of the other two referees. I am satisfied with the vet majority of their answers.

All in all, it is hard to perform a "hard experiment" and I think even harder to publish it. I strongly recommend the Nature Communications editors to publish this work, even though much hinges on modelling. We can advance 1D physics only by performing such experiments and this manuscript raises the bar even further,.

Reviewer #3 (Remarks to the Author):

In response to my comments and the ones by the other referees, the authors have deeply improved the manuscript. They have in particular added many needed details both on the experimental side and

on the modelization of the system itself.

These modifications have cleared most of the objections that I had for the previous version of the paper.

In particular the new figure 2 and figure 4, as well as the additions in the supplementary material now clarify well the regime of validity of the NLL model in connection with the experiment. The authors have also made a serious effort to address the other sources of momentum dependence of the exponent.

So although I have still some doubts on the range of validity of the yellow region in figure 4 (a range of about 2 meV on the left of the peak would mean excitations largely comparable to the bandwidth which is about 3 meV -- see the point 27 -- thus making the TLL description of excitations around EF a little bit borderline), I think that the authors have made a convincing enough case to recommend publication of the paper. The experimental results and analysis are interesting, as pointed out in my previous report, and publication of this paper could stimulate further studies that should allow to prove or disprove the interpretation of the data made in the present paper.

Response to referees

Referee comments are in blue, our responses are in black.

Reviewer 1 (Remarks to the Author):

1) The paper was revised for describing experimental procedures more clearly and additional analysis which is not directly related to the non-linear Luttinger liquid. The main discussions on the non-linear Luttinger liquid are mostly untouched.

2) As I mentioned in the first report, the experiment is significantly influenced by the disorder or else to obscure the expected momentum-dependent power law. Data in Fig. 4 is the only experimental feature that the authors believe to be the signature of the nonlinear Luttinger liquid. It is hard for me to believe the presence

of many-body effects from the nearly symmetric conductance peaks. For example, when the experimental trace (say the top trace $g(E)$ at $B = 2.66$ T for Sample A in Fig. 4) is folded along the peak position ($E = -2.2$ meV), one can find very tiny asymmetry $g(E = -2.2\text{meV} + \text{eps}) - g(E = -2.2\text{meV} - \text{eps}) < 0.1$ even at largest $\text{eps} = 1.6$ meV. It is even worse for other traces at lower B . When the data is compared with the plots for the non-interaction model, the deviation also remains 0 to 0.2 (at most) of the peak value even at the lowest energy condition. These asymmetry and deviation are comparable to the fluctuation of the experimental curve (about 10

The referee has made the confusing assumption that the peak is symmetric in the *non-interacting* case, and so is not impressed by the conductance enhancement to the left of the peaks in Figure 4. This totally misses the point that the non-interacting calculation (shown by the crosses) is significantly *asymmetric*: the conductance is lower on the left of the peak than on the right, when one carries out the full convolution of the 1D and 2D spectral functions. Once one realises this, it is clear that the enhancement of the nonlinear interacting result over this non-interacting curve is much larger than the fluctuations and uncertainties in the experimental data.

3) Generally, as the bias voltage is increased or decreased away from zero, more unwanted processes come in. This is not discussed in the paper and could be the reason for the asymmetry and deviation.

We have done the best we can to discuss and take into account unwanted processes that might enhance the conductance. We have used two different samples. We have also already considered in detail the effects of localisation by disorder in the one and a half pages of the supplementary material (section 1), where we have ruled out any effects that we can come up with. Making the experiment any clearer would be very hard, and our experiment is the state of the art. We are convinced that the enhancement is above the noise.

4) Similarly, the large disorder made the theoretical curves significantly broader. While the authors claim that the non-linear Luttinger model with disorder explains the experimental feature, this is not convincing as the accuracy of the calculation is not tested for this experiment. For example, a reference measurement that warrants the validity and accuracy may be needed to convince the readers.

We can certainly show that the typical broadening Γ is of order 0.3 meV or more, both from theoretical and experimental considerations. Theoretically, if one assumes that the 18 nm GaAs quantum wells fluctuate in width across the 2D plane by one monolayer (0.5 nm), one can calculate that their zero-point energy will vary by the order of 1 meV. This is likely to be the principal cause of the broadening Γ , though it appears to overestimate it.

Experimentally, we have obtained fits with Γ of the order of 0.5 meV in the past in various 1D-2D and in 2D-2D tunnelling measurements. Given the

constant improvements to the quality of heterostructures and in our processing techniques the values of $\Gamma = 0.2\text{ meV}$ to 0.4 meV in the latest generation of our samples are very likely and are something to be expected. We have additionally crosschecked these values of Γ by fitting the 2D-2D conductance in the parasitic region (the magenta and cyan parabolaes in Fig. 2(b)), obtaining $\Gamma = 0.25 \pm 0.03\text{ meV}$ (Sample A). This is about 20% smaller than the $\Gamma_I = 0.32 \pm 0.03\text{ meV}$ in Fig. 3(a) for the measurement with wire gates, which is natural since additional gates introduce some extra disorder. We thank the referee for asking this question and have added a corresponding remark to the text.

5) I think that the paper is worthy for publication in some journal if the interpretation is made more speculative. However, I do not recommend this paper for publication in Nature Communications.

We hope the above comments have adequately refuted the referee's criticisms of the work.

Reviewer 2 (Remarks to the Author):

6) First, I commend the authors for this well executed experiment. While I am still not clear the manuscript is "clear", I recommend it for publication in Nature Communications. There are times when we are "ahead" in research that is not easy to render our work pedagogical. But still, it is our duty.

7) As a competitor in the field, I commend the authors for having undertaken this experiment. As I said earlier, there appears to be an effect but not a smoking gun. I also commend the authors for their due diligence regarding my criticisms, and those of the other two referees. I am satisfied with the vet majority of their answers.

8) All in all, it is hard to perform a "hard experiment" and I think even harder to publish it. I strongly recommend the Nature Communications editors to publish this work, even though much hinges on modelling. We can advance 1D physics only by performing such experiments and this manuscript raises the bar even further.

We thank the referee for their positive comments.

Reviewer 3 (Remarks to the Author):

9) In response to my comments and the ones by the other referees, the authors have deeply improved the manuscript. They have in particular added many needed details both on the experimental side and on the modelization of the system itself.

10) These modifications have cleared most of the objections that I had for the previous version of the paper.

11) In particular the new figure 2 and figure 4, as well as the additions in the supplementary material now clarify well the regime of validity of the NLL model in connection with the experiment. The authors have also made a serious effort to address the other sources of momentum dependence of the exponent.

12) So although I have still some doubts on the range of validity of the yellow region in figure 4 (a range of about 2 meV on the left of the peak would mean excitations largely comparable to the bandwidth which is about 3 meV – see the point 27 – thus making the TLL description of excitations around E_F a little bit borderline), I think that the authors have made a convincing enough case to recommend publication of the paper. The experimental results and analysis are interesting, as pointed out in my previous report, and publication of this paper could stimulate further studies that should allow to prove or disprove the interpretation of the data made in the present paper.

We are pleased that the referee finds our work worthy of publication in Nature Communication. As a comment on the range of validity (the yellow region), nLL is a new model and its exact limits of applicability even within the theory are not well established yet. It might be tempting to just linearly translate the range of validity of the linear LL to the nonlinear model, since linear LL is a part of the nLL. However, there are other considerations:

Bethe-ansatz calculations for the model of spinless fermions in [PRB 93, 075147 (2016)] show that the nonlinear power-law is still visible at least at distances of $\lesssim 0.5E_F$ from the spectral threshold (parabola in the hole sector). The same calculation shows the applicability range of the linear LL of $0.1E_F$ from the Fermi point. Both numbers are for reasonably strong interaction strengths and were obtained by visual inspection of the numerical results of Bethe ansatz calculations of the spectral function. While the details of the spinful fermion model (there is no calculation for it so far) should change these numbers somewhat, they look consistent with what we observe in this experiment.

In this experiment we measure conductance, which is a convolution of the 2DEG's spectral function with that of the interacting 1D system. Since the projection of the 2DEG's spectral function on 1D is not a delta function (it is $\theta(E_F - E)/\sqrt{E_F - E}$), visibility of the nonlinear power-law in conductance can

be extended in the energy domain. This effect would not manifest itself too much around E_F since the spectral functions of LL and nLL are finite and are of the same order. But for the nonlinear power laws there are no extra features around them far into the nonlinear regime—the spectral function beyond the power-law singularity is much weaker. Thus, we might be able to continue to see the apparent power law in conductance further from the threshold.

These arguments indicate that the currently chosen width of the yellow region is reasonable. But without any doubt this issue deserves more detailed studies and we sincerely hope it will be done in the future by other groups or by us. In order to increase readability of the manuscript we keep this discussion to a minimum. Our approach for choosing the extent of the yellow region is phenomenological: we select its width as the range where the interacting model still gives a reasonable fit. We have added a corresponding remark to the manuscript.